# Evaluating topological fitness of human brain-inspired sub-circuits in Echo State Networks

## Abstract

Recent years have witnessed an emerging trend in neuromorphic computing that centers around the use of brain connectomics as a blueprint for artificial neural networks. Connectomics-based neuromorphic computing has primarily focused on embedding human brain large-scale structural connectomes (SCs), as estimated from diffusion Magnetic Resonance Imaging (dMRI) modality, to echo-state networks (ESNs). A critical step in ESN embedding requires pre-determined read-in and read-out layers constructed by the induced subgraphs (e.g., *a priori* set of functional sub-circuits/networks) of the embedded reservoir (e.g., SCs). As *a priori* set of functional sub-circuits are derived from functional MRI (fMRI) modality, it is unknown, till this point, whether the embedding of fMRI-induced sub-circuits/networks onto SCs is well justified from i) the neuro-physiological perspective and ii) ESN performance across a variety of tasks. In this paper, we proposed a pipeline to implement and evaluate ESNs with various embedded topology and processing/memorization tasks. To this end, we showed that different performance optimums are highly dependent on the neuro-physiological characteristics of these pre-determined fMRI-induced sub-circuits. In general, fMRI-induced sub-circuit-embedded ESN outperforms simple bipartite and various null models with feed-forward properties commonly seen in MLP for different tasks and reservoir criticality conditions. Noticeably, we found that the reservoir model performance is heavily dependent on the functional sub-circuits neuro-physiological properties with respect to different cognitive tasks and their corresponding computation-memorization balances. Specifically, we showed that default mode network's superior performance across the majority of tasks is related to its functional dichotomy property. Finally, we provided a thorough analysis of the topological properties of pre-determined fMRI-induced sub-circuits and highlighted their graph-theoretical properties that play significant roles in determining the ESN performance.

## 1 Introduction

One of the prominent directions in building high-performance or general intelligence artificial systems is implementing biologically plausible models, particularly using human brain connectomes as the underlying graph of neural networks (Damicelli et al. (2022); Suarez et al. (2020); McDaniel et al. (2022); Suarez et al. (2023)). Most state-of-the-art biologically inspired artificial connectomes are realized as reservoir networks with structural human brain data. However, structural connectomes (SC) derived from diffusion Magnetic Resonance Imaging (MRI) are generally rigid in both topology and scale compared to functional connectomes (FC) from functional MRI data. Thus, in this work, we explore the prospect of applying functional connectomes to reservoir Echo-state Networks (ESNs) under various conditions to comprehensively evaluate model performance, as well as build a consistent framework for implementations of connectomic data in artificial networks.

Motivated by the fact that the topology of a neural network has a tangible impact on its performance, and that complex network topology sometimes improves the performance of the model, we systematically explore more complex and biologically inspired topology in reservoir neural networks. A critical step in ESN embedding requires pre-determined read-in and read-out layers with inputs

sampled from the induced subgraphs of the embedded reservoir. Typically, *a priori* set of functional networks (FNs), such as Yeo's FNs (Yeo et al. (2011)), is used as pre-determined subgraphs for read-in and read-out input selections. Since *a priori* set of FNs are derived from functional MRIs, yet, these pre-determined sub-circuits are mapped onto SC, it is unknown whether FN-to-SC embedding is well justified from i) neuro-physiological perspective and ii) ESN performance.

In this paper, our contributions are as follows: i) We proposed a pipeline to implement and evaluate echo-state reservoir networks with various embedded topology on various processing and memorization tasks. To this end, we tested various topological embedding and evaluated their respective performance on diverse tasks, showing different performance optimums are highly dependent on the embedded neuro-physiological architecture of *a priori* functional networks and the corresponding task configurations; ii) we show that, in general, complex topology perform better than simple bipartite models or null models with feed-forward properties commonly seen in MLP for certain tasks and reservoir criticality conditions. As such, we also found that, contrary to earlier literature, the performance of the reservoir model is not strictly defined by the reservoir's echo-state property defined by the spectral radius of the connectivity matrix; iii) we found that there exists a differentiated performance of functional networks induced from the human structural connectome with respect to different cognitive tasks and their corresponding computation-memorization balances. We analyzed the topological properties of *a priori* functional networks and highlighted their graph-theoretical properties that play significant roles in determining the final performance of the echo-state model.

## 2 BACKGROUND AND RELATED WORKS

### 2.1 TOPOLOGICALLY-EMBEDDED ARTIFICIAL NEURAL NETWORK AND ECHO-STATE-NETWORKS (ESN)

Echo-state networks are a variant of the classical reservoir computing (RC) paradigms, where the reservoir is a recurrent layer of the classical recurrent neural network (Verstraeten et al. (2007)). RCs are typically proposed in place of regular RNN variants as a lightweight model, where the hidden states are high dimensions while only the readout layer weights of the model must be updated during training. The frozen nature of the reservoir layer allows any topology and unit dynamics to be embedded. As a result, RCs have been implemented as predictors of chaotic dynamics in physics simulators, or used as a time-series prediction model in general (Chattopadhyay et al. (2020); Shahi et al. (2022); Bompas et al. (2020); Huhn & Magri (2022); Platt et al. (2022)). ESNs model time as discrete steps, making them suitable for discrete-time sampling data seen in real-world time-series datasets (Cucchi et al. (2022)).

In the current literature, the relationship between the topological structure of an NN and its performance is still unclear. While complex networks may yield higher predictive performance or parameter efficiency (Kaviani & Sohn (2021)), it is difficult to generalize randomly wired models outside the investigated context. Under the constraint of MLP-random interpolation, several random network models rewired similarly to the small-world regimes perform well on several real-life problems (Erkaymaz et al. (2017); Erkaymaz & Ozer (2016)). Different topology perform well in limited tasks on small network sizes enabled by the learning matrix sequential algorithm for training directed acyclic graphs (DAGs). Random graph topology are also shown to perform well as image classifiers on the ImageNet dataset, especially the Watts-Strogatz (WS) model of small-world networks (Xie et al. (2019)). Boccato et al. (2024) echoes the statement on the performance of small-world models in the case of synthetic data, albeit the difference between WS and other complex graph families is less pronounced, possibly owing to the sufficient degree of topological complexity for function representation.

### 2.2 BRAIN CONNECTOMICS AND HUMAN BRAIN FUNCTIONAL SUB-CIRCUITS

Human Brain connectomics studies how the human brain is **structurally connected** and how a heterogeneous repertoire of resting-state or task-related **functional circuits** emerge on top of it. There are two types of mesoscopic structures in large-scale brain networks: localized and non-localized. In this paper, we focus on investigating localized mesoscopic structures which are sub-systems learned from local/quasi-local network properties such as brain regions of interest (ROIs) or functional edges, or correlations among neighboring nodes (Duong-Tran et al. (2024)). In brain

connectomics, these sub-structures are induced from a wide array of techniques, including but not limited to clustering (Yeo et al. (2011); Power et al. (2011)) or low-dimensional approximation of high-dimensional dynamics (Shine et al. (2016); Shine & Poldrack (2017); Shine et al. (2019)). The most commonly known localized mesoscopic structures in brain networks are often referred to as functional sub-circuits or functional networks (FNs) (Yeo et al. (2011); Sporns & Betzel (2016); Duong-Tran et al. (2019; 2024)). From here on, we will use the two terms functional sub-circuits and FNs interchangeably.

### 2.3 HUMAN-BRAIN-CONNECTOMICS ESNS

In the context of modeling the human brain connectome, ESNs are implemented as prototype models of the structural brain topology. Suarez et al. (2020) shows that human brain connectivity captured through diffusion imaging performs better than random null network models in the critical regime, while also demonstrating computationally relevant properties of resting-state functional brain network parcellation. Given a minimum level of randomness and connection diversity from the original structural connectome, biologically inspired reservoir networks' performance matches that of any classical random network model (Damicelli et al. (2022)). The authors of d'Andrea et al. (2022) conclude that the modular structure of reservoirs significantly impacts prediction error among multiple other features. While the connections are modeled in a recurrent network rather than strictly an RC, Achterberg et al. (2023) shows that training an RNN constrained by reservoir features such as spatial wiring cost and neuron communicability leads to the emergence of structural and functional features commonly found in primate cerebral cortices.

## 3 ANALYZING CONNECTOME-BASED RESERVOIR MODEL

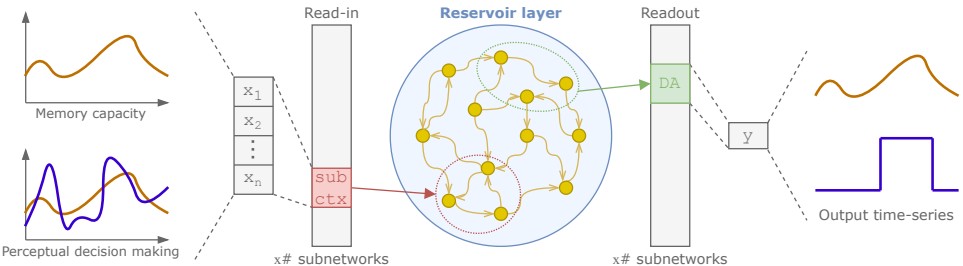

Figure 1: Overview of the pipeline. Memory, processing, or mixed datasets are fed into the model through a static input layer, which is then projected into higher dimensions with the reservoir layer, and readout using a ridge regression output layer. Structural, functional, and null model connectomes are embedded in the reservoir layer.

### 3.1 PRELIMINARY: ECHO-STATE IMPLEMENTATION

To examine different topologies as computational graphs, we utilized the implementation of the classical ESN (Jaeger (2001)) from the `conn2res` package (Suarez et al. (2023)) which has the following form:

$$x[t] = \tanh\left(\mathbf{W}^R x[t-1] + \mathbf{W}^I u[t]\right) \tag{1}$$

$$z[t] = \mathbf{W}^O x[t] \tag{2}$$

The reservoir state $x[t]$, data input $u[t]$, and reservoir readout output $x_{out}[t]$ are, respectively, real value vectors of $N_r$, $N_i$, and $N_o$ dimensions. Matrices $\mathbf{W}^R$, $\mathbf{W}^I$, and $\mathbf{W}^O$ are the recurrent reservoir internal weight matrix, the reservoir input weight matrix, and the readout weight matrix, respectively. Through experimentation, we chose the tangent function $\tanh$ as the nonlinearity for the entire reservoir. The reservoir is initialized in the origin, i.e., $x[0] = 0$. We initialize input matrix $\mathbf{W}^I \in \mathbb{R}^{n \times m}$ ($n$ is the number of input nodes, $m$ is the input feature length):

$$\mathbf{W}_{ij}^I = \begin{cases} C, & \text{if } i = j \mod n \\ 0, & \text{otherwise,} \end{cases} \tag{3}$$

where $C$ is a predetermined input factoring constant, identical for all input features and read-in nodes.

We train the ESN by optimizing the readout matrix $\mathbf{W}^O$, solving the linear regression problem $y[t] = \mathbf{W}^O x[t]$ using the training data $\{u[t], y[t]\}_{t=1,...,T}$. The matrix $\mathbf{W}^O$ is optimized using *ridge regression* through the formula:

$$\mathbf{W}^O = \mathbf{Y}\mathbf{X}^\top(\mathbf{X}\mathbf{X}^\top + \lambda \mathbf{I})^{-1}, \tag{4}$$

where $\mathbf{X} \in \mathbb{R}^{N_p \times T}$ is the matrix containing temporal reservoir states $x[t]$ of a certain $N_p$ nodes ($N_p < N_r$) of the reservoir computed from the previous states and the input $u[t]$ for $t = 1,...,T$, $\mathbf{Y} \in \mathbb{R}^{N_o \times T}$ is the ground-truth matrix containing $y[t]$, $\mathbf{I} \in \mathbb{R}^{N_p \times N_p}$ is the identity matrix, and $\lambda$ is the regularization parameter. Additionally, we keep track of the spectral radius $\alpha$ of the reservoir's adjacency matrix to examine the reservoir's adherence to the echo-state property (ESP), a condition quantifying the reservoir's unique stable input-driven dynamics (Jaeger (2001)).

## 3.2 Pipeline and structural control model

**Reservoir initialization** There are two components in the original structural connectome dataset: topology (wiring between regions of the brain regardless of connection strength) and weights (precise connection strengths between mesoscale regions). The adjacency matrix $\mathbf{A}$ of each connectome contains both components, where $\mathbf{A}$ is symmetric by constraint of the original imaging method and the ESN model (Figure 1). The connectome connectivity and edge weights are embedded in the reservoir as an undirected graph, where $\mathbf{W}^R$ is constructed from $\mathbf{A}$ by min-max scaling $\mathbf{A}$ to range $[0, 1]$ and normalizing $\mathbf{A}$ by the spectral radius constraint $\alpha$:

$$\mathbf{W}^R = \alpha \frac{\mathbf{A}_0}{\rho(\mathbf{A}_0)} \tag{5}$$

where $\mathbf{A}_0$ is the scaled weighted connectivity matrix, and $\rho(\mathbf{A}_0)$ is the spectral radius of $\mathbf{A}_0$.

**Read-in and read-out functional networks** The reservoir network is generally partitioned using intrinsic functional networks, a connectivity-based partition of brain network into functionally similar groups of areas (Wig (2017)). The reservoir input weight matrix $\mathbf{W}^I \in \mathbb{R}^{n \times m}$ route the input signals to only certain $m$ nodes in the reservoir (Figure 1, 2). The input nodes are chosen as an entire functional network to examine message propagation on the reservoir computational graph; in this paper, we chose subcortical regions as input nodes, owning to the plausibility of these regions serving as relay stations for incoming sensory signals. Each of the other seven cortical regions is chosen as the read-out node-set, where the readout weight matrix $\mathbf{W}^O$ connects the $N_p$ nodes of the functional network to the readout module.

**Structural control** We embed the structural human brain connectome into the reservoir as the default topology map. Consensus adjacency matrices are constructed from the original connectomes with bootstrap resampling to provide a reliable estimate of the model performance. The structural connectome is grouped into seven intrinsic functional networks and one subcortical region according to the Yeo functional mapping and parcellation (Thomas Yeo et al. (2011)). These structures are known as sub-circuits or FNs, previously defined in Section **2.2**.

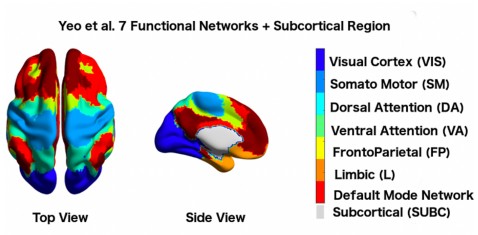

Figure 2: Human brain functional networks (sub-circuits) as parcellated by Yeo and colleagues Yeo et al. (2011).

### 3.3 Variants: Complex graph null models

In general, the models described in this section aim to represent various aspects of the biological mesoscale complex networks and examine the different components of structural brain connectomes, mainly through null models that ablate one or multiple properties of the original connectomes.

**Subgraph** To examine the message diffusion in the reservoir, we constructed an induced subgraph partitioning of the original graph and extracted only the subcortical input and current cortical output regions for the reservoir model. The graph in the reservoir is thus an *vertex-induced subgraph* of only the input-output networks of the original graph; for example, if the current output network is VIS, then the reservoir adjacency matrix is constructed from the original graph $G$ by $G[V(\text{subctx}) \cup V(\text{VIS})]$.

**Maslov-Sneppen (MS) rewire** We preserve the original degree-sequence of the structural connectome and rewired each edge in the connectome exactly once using the Maslov-Sneppen algorithm (Maslov & Sneppen (2002)). From the rewiring procedure, we obtained a reference null connectome with the original constraint still moderately enforced.

**Uniform weights** We preserved the underlying topology of the connectome while randomizing the connection weights between nodes of the network. The connecting edges between all nodes are randomized and sampled from the uniform distribution $U(0, 1)$, while the adjacency matrix is kept symmetric.

**Bipartite null** Using the original node counts from the subcortical region and the current output region as baseline (i.e., the node set of the subgraph model), a complete undirected bipartite graph consisting of only the input and output node sets is created. The weights between the two node sets are sampled from the uniform distribution $U(0, 1)$, while the adjacency matrix is symmetric. The bipartite model is analogous to an untrained two-hidden-layer feedforward network model, where the first layer is of the same size as the subcortical nodes and the second as the output functional network (e.g., VIS).

**Newman configuration model** An entirely new graph is constructed from the degree sequence of the original structural connectome, as opposed to only rewiring each edge once. The configuration model generates a random undirected pseudograph by randomly assigning edges to match a given degree sequence (Newman (2003)). Self-loops are then removed from the existing graph to obtain the symmetric null connectome adjacency matrix for the reservoir.

## 4 Experimentation: Connectome-dependent Performance of ESNs

### 4.1 Data

**Structural and functional connectomes** Adapted from structural consensus connectomes from (Suarez et al. (2020)). The connectome is divided into 463 or 1015 approximately equally sized nodes. A group-consensus approach is adopted to mitigate reconstruction inconsistencies (Betzel et al. (2019)) and network measure sensitivities between different maps. Functional connectomes are obtained from a publicly available dataset (Derived Products from HCP-YA fMRI, Tipnis et al. (2021)). More information on the brain connectomic datasets is available in Appendix A.1.

**Synthetic sequential behavioral data** We used synthetic behavioral data to simulate computation from multiple signal sources and assess the memory capacity of the reservoir under various means. The PerceptualDecisionMaking (PDM) (Britten et al. (1992)) task is a two-alternative forced choice task requiring the model to integrate and compare the average of two stimuli, with the PerceptualDecisionMakingDelayResponse (PDMDR) (Inagaki et al. (2019)) variant artificially inducing a memory requirement by prompting the model to answer after a random delay. The context-dependent decision-making task ContextDecisionMaking (CDM) (Mante et al. (2013)) requires the model to make a perceptual decision based on only one of the two stimulus inputs from two different modalities, where a rule signal indicates the relevant modality. The assessment for the memory capacity

of the model is the MemoryCapacity (MemCap) (Suarez et al. (2020)) dataset, where the encoding capacity of the model and readout modules are estimated through sequential recall of a delayed time-series signal under various time lags, quantified by the Pearson correlation between the target sequence and the model's predicted memorization signal.

## 4.2 EXPERIMENTS

**Procedure, Setup, and Benchmarking**   The general procedure and pipeline for all network models are described in Section 3. Each model is evaluated on 7 experiments each according to the 7 readout Yeo networks, with the synthetic datasets identically generated for all functional networks on each of the same 1000 tests. The models are primarily evaluated on F1-score (for PDM, PDMDR, CDM) and Pearson correlation (for MemCap). Details on the experimental setup and implementation libraries are included in the Appendix B.

## 4.3 EMPIRICAL PERFORMANCE: PROCESSING AND MEMORIZATION

We evaluate the performance of structural and null models under the mentioned processing and memorization synthetic tasks, showing the performance of each model over multiple reservoir spectral radius (alpha) in Figure 3. Models under perceptual and memorization tasks show performance decay from $\alpha = 1$, while contextual models perform well even in the chaotic $\alpha > 1$ regime. The memory capacity of reservoirs shows the most notable decline in the chaotic regime, similar to results obtained in Suarez et al. (2020), and all models perform better than the fully connected bipartite null counterpart with the same read-in and read-out node count, showing gains in having higher topological complexity when strictly compared to 2-layer feedforward model. Model performance across 1000 different instantiations of the reservoir is fairly stable, showing that degree-conserving null models and bipartite model performed worse than the original biological structural model, structure-based subgraph model, and uniformly randomized structural weight model at close to criticality $\alpha = 0.95$ (Figure 4).

Between different complex topologies aside from the bipartite models, models that preserve the original structural brain topologies perform better than other null models. The trend is consistent across all alpha values except for CDM, where higher alpha shows the null uniform weights model with identical topology outperforming the original structural model. Aside from MemCap, control structural and subgraph networks perform almost identically, suggesting that information does not propagate through non-input-output pathways in the reservoir for three NeuroGym tasks. MS rewire and configuration model with only the degree sequence preserved show worse performance than the nulls without topological randomization, suggesting that the topological control models contain features that lead to better processing and memorization performance.

## 4.4 READ-OUT PERFORMANCE FROM FUNCTIONAL NETWORKS

We look into null models in comparison to the original structural model to compare the performance when each functional network is chosen as the read-out node-set. The models under consideration are the structural control model, the MS one-step model, uniform weights null models, and the label permutation null, all under the equivalently-performing $\alpha = 0.95$ (Figure 6). We observed that for 3 out of 4 datasets (PDMDR, MemCap, CDM) under the control model, the default mode network (DMN) and somatomotor network (SM) perform better than the other functional networks, with the DMN significantly outperforming SM in the majority of cases. This trend persists in other null models other than control for the mentioned 3 tasks, with the ranking maintained in PDMDR and MemCap, and DMN traded place with SM in CDM in nulls other than permutation. PDM shows a contrastive difference to the other tasks, where the underperforming dorsal attention (DA) and frontoparietal (FP) networks perform better than the other networks, while DMN and SM notably performed worse than the other subnetworks. The performance ranking between different read-out functional networks for PDM strongly persisted in other null models, regardless of weighting or structural randomization.

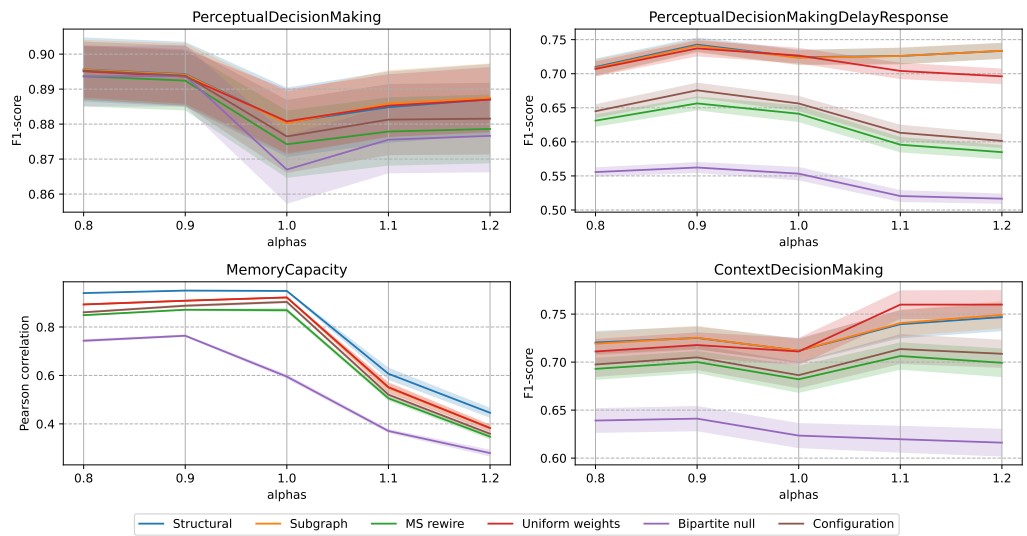

Figure 3: General performance of various reservoir models tested on four datasets. Performance is shown as a function of the reservoir's spectral radius alpha.

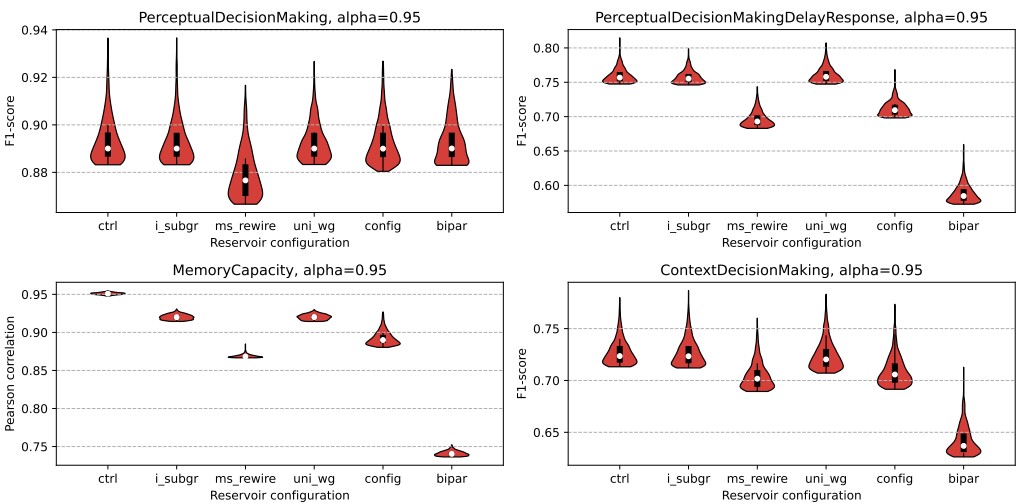

Figure 4: Performance of various reservoir models on four datasets w.r.t. the reservoir configuration at spectral radius alpha near criticality (0.95), demonstrating the best performances of ESNs in most cases. (config: degree-sequence-preserving configuration random graph, bipar: bipartite random graph)

### 4.5 COMPARISON WITH OTHER RNNS AND TESTING WITH REAL-WORLD DATA

**Testing with traditional RNN models** (Figure 5) We benchmarked the ESN variants against LSTM and RNN models of various sizes. We found that sizes smaller than the tested configurations do not perform as well as the shown ones, regardless of bi-directionality. In general, traditional RNNs perform reasonably well but are not as well-rounded as ESNs and are also highly unstable in training environments.

**Testing of echo-state models in real-world time series prediction tasks** (Table 1) We present the ESN model, a control variant using the structural connectomes of size 500 as described in the paper, for predicting the course of COVID-19 spread. We obtained the datasets from open sources:

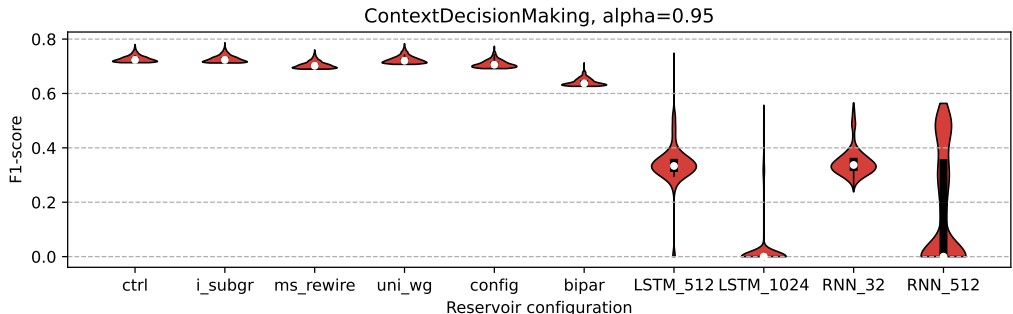

Figure 5: F1-score performance of various reservoir models on three datasets w.r.t. the reservoir configuration at spectral radius 0.95. ESN configurations are identical to the original paper's description, LSTM and RNN models are followed by the size of the hidden layer.

| Model | Up to next 7 Days | | | Up to next 14 Days | | | Up to next 21 Days | | |
|---|---|---|---|---|---|---|---|---|---|
| | England | France | Spain | England | France | Spain | England | France | Spain |
| GAUSSIAN_REG | 14.13 | 4.44 | 51.28 | 12.00 | 3.03 | 43.12 | 10.04 | 1.96 | 31.58 |
| RAND_FOREST | 7.16 | 5.01 | 37.05 | 10.01 | 7.13 | 51.72 | 13.03 | 9.76 | 61.38 |
| PROPHET | 14.45 | 13.86 | 75.86 | 23.43 | 21.25 | 114.87 | 33.59 | 27.88 | 149.51 |
| ARIMA | 9.51 | 9.08 | 40.54 | 9.63 | 8.78 | 48.46 | 9.77 | 8.13 | 56.45 |
| Bi-LSTM | 8.20 | 6.12 | 42.64 | 7.86 | 8.47 | 36.45 | 7.09 | 8.94 | 35.73 |
| **ESN (control)** | 13.34 | 4.41 | 48.95 | 11.24 | 3.01 | 42.60 | 9.35 | 1.95 | 33.63 |

Table 1: Performance of all experimental model evaluated based on Mean Absolute Error, prediction of the number of COVID-19 new cases in England, France, and Spain in the next 3-21 days.

England, France, and Spain. For each model, we performed rolling-window training and testing, where each time the model is evaluated it is trained on historical data up to the number of days (3-21 days later) in advance to be predicted until the target time point, then the model is evaluated from the target time point to the number of days in advance to be predicted. We measured each model's performance using Mean Absolute Error.

The models tested follow the same notation described in prior work on COVID-19 prediction (Panagopoulos et al. (2021)). The additional models we tested are briefly described in Appendix B.4.

In general, we found that the performance of our echo-state models is well-rounded and fairly competitive w.r.t. other benchmarking models, while also offering the advantage of being faster to optimize when compared to trained network models (i.e., LSTMs), or being more generous in terms of hyperparameter optimization compared to specialized time-series model (e.g., ARIMA).

## 5 FUNCTIONAL SUB-CIRCUITS' TOPOLOGICAL PERFORMANCE ANALYSIS

In this section, we analyze the neuro-physiological characteristics of functional networks from a topological perspective and their corresponding contributions to better model performance than others. Specifically, we analyze FN's graph-theoretical measures via two viewpoints: i) statics (e.g., size, betweenness, modularity) and dynamics (e.g., communicability) properties, quantifying their relationship with model performance. All network measures used in this section are described in details in Appendix C.

### 5.1 READ-OUT FUNCTIONAL NETWORKS' STATICS PROPERTIES

In this sub-section, we investigate FN statics properties through their node count statistics and betweenness measures (Puzis et al. (2007)). Based on Figure 5, we see that there exists an association, for some FNs, between their sizes (through node count statistics) and their betweenness scores. Specifically, DMN, the largest sub-circuits, also has the highest betweenness score. More impor-

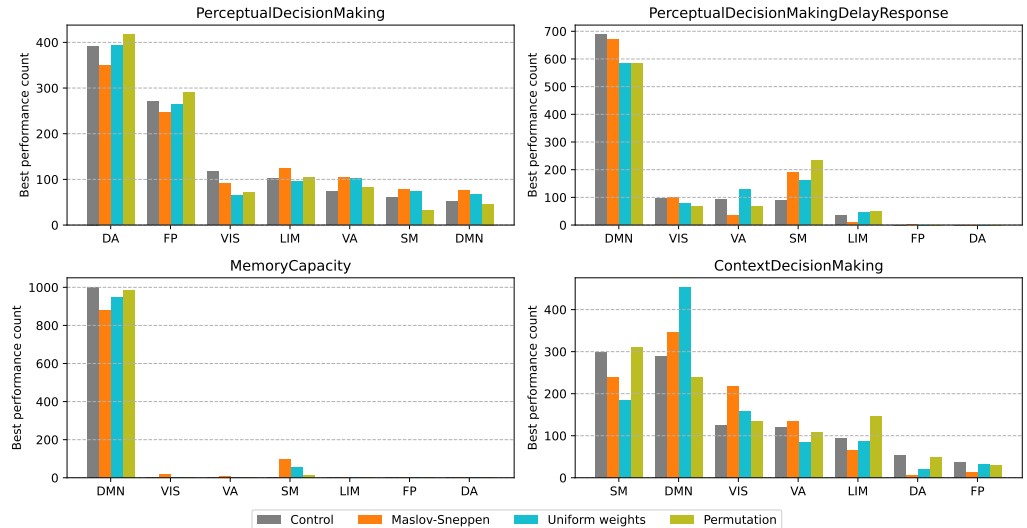

Figure 6: The number of best performance runs by model and nulls across 1000 runs, by each functional network on identical initialization of each dataset. Note that the total of all functional networks from the same color (reservoir configuration) adds up to exactly 1000.

tantly, there exists an association between the number of best performance analyses based on FN (Figure 4) to their betweenness scores. Specifically, DMN, ranked top 2 in 3 out of 4 tasks per Figure 4, ranked first in betweenness score. Other "big" sub-circuits such as VIS or SM also have competitive betweenness.

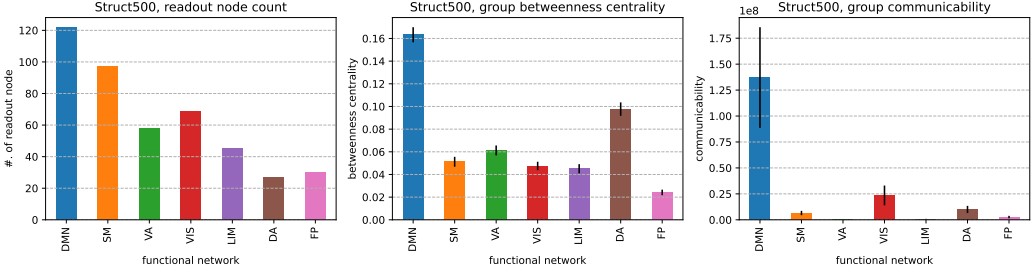

Figure 7: Struct500 (463 nodes) connectome, various graph statistics, 1000 different consensuses.

## 5.2 READ-OUT FUNCTIONAL NETWORKS' DYNAMICS PROPERTIES

Here, we evaluate read-out FN's dynamics through diffusivity and communicability perspectives, measuring the extent to which information flows through the FN topology (Estrada & Hatano (2008)). Noticeably, we found that the communicability score reaffirms DMN dominance, not only in the statics domain (e.g., betweenness) but also in the dynamics domain (e.g., communicability). Nonetheless, DMN dominance is significantly larger (five times larger than the runner-up: VIS). This result would further strengthen the anticipated best performance from DMN for 3 out of 4 tasks.

In Figure 8, we analyze the correlation between graph statistics and empirical performance on PDMDR. Communicability spread shows that functional networks with high communicability generally perform well compared to networks with lower communicability, while there is no significant correlation between modularity and performance. Importantly, group betweenness centrality of functional networks is separated into four separate clusters, with certain networks or ranges of centrality performing better than the others (third panel, Figure 8). For other datasets (results in Ap-

pendix D) the results are generally the same for communicability and modularity, while betweenness centrality clusters are generally the same with CDM and MemCap reproducing the performance seen in PDMDR.

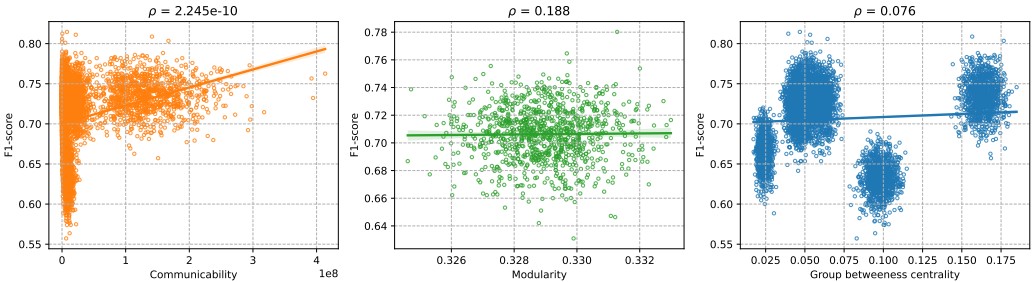

Figure 8: Structural 500 nodes, statistic-performance correlations, PDMDR, $\alpha = 0.95$, with slope of best linear fit $\rho$.

### 5.3 TOPOLOGICAL FITNESS OF A PRIORI SET OF FNs ON CONNECTOME-BASED ESN

In this section, we propose the concept of FN topological fitness for connectome-based ESN using graph modularity score (Clauset et al. (2004a)). To evaluate the topological fitness of an *a priori* set of FN, we measure Q with pre-set $\sigma$ (e.g., Yeo's FNs Yeo et al. (2011)). We performed cross-comparison between processed functional and structural connectome on modularity measures, performing community detection and comparing adjusted mutual info score with the original parcellations. We found that, despite the functional networks parcellation derived from functional connectomes, the modularity of the structural topology is higher than functional (Figure 9), while structural connectomes also perform better than functional (Appendix D). We thus hypothesize that Yeo's networks derived from Lausanne parcellation (Suarez et al. (2020)) and Schaefer parcellation (Tipnis et al. (2021)) are not information-theoretically aligned, or that topological fitness strongly determines model performance.

## 6 DISCUSSION AND CONCLUSIONS

In this study, we explored bio-plausible topology as computational structures in reservoir neural networks, particularly ESNs, and examined whether complex functional topologies contribute to performance in recurrent network settings. We found that different topological properties are conducive to the processing of different tasks. The topology of default mode network (DMN) facilitates better processing and memorization, implying a link to the "functional dichotomy" of DMN found in traditional neuroscience studies (Uddin et al. (2009)).

Functional networks are embedded and extensively tested in reservoir echo-state networks where different performance scalings and dynamic entry points are evaluated. In general, bigger connectomes per-

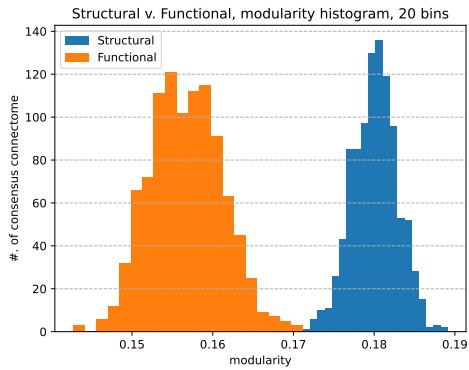

Figure 9: Modularity histogram: difference between structural and functional connectome.

form better albeit with diminishing returns, and the flow through functional networks determines how well the model performs. Interestingly, optimal thresholds for task performance are slightly higher than positive. Experiments have also shown that the model's performance is somewhat dependent on the absolute size of the readout functional networks. Further experimentations on integrating functional signals to embedded reservoirs will be conducted to optimize the use of rs-fMRI or task fMRI data, more closely observing model performance and computational efficiency and expanding the model representation space using different activations.

**Ethics Statement.** Given the nature of the work, we do not foresee any negative societal and ethical impacts of our work.

**Reproducibility Statement.** Source codes for our experiments are provided in the supplementary materials of the paper. The details of our experimental settings are given in Section 4 and the Appendix B. All datasets that we used in the paper are published, and they are easy to access in the Internet.

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

## A ADDITIONAL DATA DESCRIPTION

### A.1 RESERVOIR CONNECTOMIC

For structural connectomes, we used connectomes from Suarez et al. (2020), publicly available at `https://zenodo.org/records/4776453`, which was originally processed from the University of Lausanne's data, also available at: `https://zenodo.org/records/2872624`. For functional connectomes, we obtained the parcellated connectomes from the UCSD public repository: `https://library.ucsd.edu/dc/object/bb59818382`. All data are available for public use with proper citations (included in the main text). None of the authors of this paper overlapped with the authors of the published datasets mentioned.

### A.2 COVID-19 DATA

The ground truth for number of confirmed COVID-19 cases per region is gathered for four regions through open data:

- Italy: https://github.com/pcm-dpc/COVID-19/blob/master/dati-province/dpc-covid19-ita-province.csv
- England: https://coronavirus.data.gov.uk/
- France: https://www.data.gouv.fr/en/datasets/donnees-relatives-aux-tests-de-depistage-de-covid-19-realises-en-laboratoire-de-ville/
- Spain: https://code.montera34.com:4443/numeroteca/covid19/

We directly used the preprocessed final version of the data from Panagopoulos et al. (2021) GitHub repository in each country's subfolder in the data folder, publicly available at: `https://github.com/geopanag/pandemic_tgnn/tree/master/data`.

## B DETAILED EXPERIMENT SETTINGS

### B.1 FRAMEWORK

All of the code provided is in Python, with primary experiments run in `conn2res` (Suarez et al. (2023)) framework for reservoir computing. The specific implementation of `conn2res` that we used is publicly available at `https://github.com/netneurolab/conn2res`. Detailed implementation is included in the supplementary material as .zip file.

### B.2 HARDWARE

All experiments were conducted on a single-CPU server with no GPU, with the following specifications:

- CPU: 1x Ryzen 9 7950x
- RAM: 64GB DDR5 4800MT/s

### B.3 HYPERPARAMETERS

We provide the hyperparameters for our experiments w.r.t. the `conn2res` (Suarez et al. (2023)) framework. Our attached supplemental code also contains all original hyperparameters used in experiments.

- Number of experiment runs $N = 1000$
- Reservoir input factor $C = 0.001$
- Train-test split: 80:20
- Number of NeuroGymTask trials: 1000 (PDM, PDMDR, CDM)
- Number of MemCap trials: 4050

- Number of reservoir nodes (structural): 463 (450 cortical nodes + 13 subcortical nodes)
- Ridge regressor hyperparameters: alpha=0.5, fit_intercept=False

Hyperparameters for traditional RNNs models are identical to main experiments. For full details on hyperparameters used, refer to attached supplemental code.

### B.4 COVID-19 BENCHMARK MODELS

The models tested follow the same notation described in prior work on COVID-19 prediction (Panagopoulos et al. (2021)). The additional models we tested are:

- GAUSSIAN_REG: Gaussian Process Regression non-parametric model implementing Gaussian processes (Ketu & Mishra (2021), `https://scikit-learn.org/stable/modules/generated/sklearn.gaussian_process.GaussianProcessRegressor.html`).
- RAND_FOREST: Random Forest Regression case prediction model based on decision tress (Galasso et al. (2022), `https://scikit-learn.org/stable/modules/generated/sklearn.ensemble.RandomForestRegressor.html`).
- Bi-LSTM: Same as paper, but tested with a two-layer bidirectional LSTM that retains the sequence of cases for the last 7 days.

## C  NETWORK MEASURES

### C.1  BETWEENNESS CENTRALITY

Betweenness centrality is a measure of information routing through a network, where the shortest-path betweenness centrality is computed for nodes (Freeman (1977)). Betweenness centrality $c_B$ of a node $v$ is the sum of the fraction of all-pairs shortest paths that pass through $v$ (Brandes (2001)):

$$c_B(v) = \sum_{s,t \in V} \frac{\sigma(s,t|v)}{\sigma(s,t)}$$

where $V$ is the set of all nodes, $\sigma(s,t)$ is the number of shortest $(s,t)$-paths, and $\sigma(s,t|v)$ is the number of those paths passing through some node $v$ other than $s,t$. If $s = t$, $\sigma(s,t) = 1$; if $v \in s, t$, $\sigma(s,t|v) = 0$.

### C.2  COMMUNICABILITY

Communicability is a measure modeling the extent of information transfer between nodes in the network, effectively provides a way to quantify the extent to which two regions in a network can communicate with each other (Estrada & Hatano (2008)). Communicability is computed using a spectral decomposition of the adjacency matrix. The communicability $C$ between two nodes $u$ and $v$ is computed using the connection between powers of the adjacency matrix and the number of walks:

$$C(u,v) = \sum_{j=1}^{n} \phi_j(u)\phi_j(v)e^{\lambda_j}$$

where $\phi_j(u)$ is the $u^{\text{th}}$ element of the $j^{\text{th}}$ orthonormal eigenvector of the adjacency matrix with eigenvalue $\lambda_j$

### C.3  MODULARITY

Modularity is a measure of the degree to which the network can be separated into clearly independent groups. As previously defined and reduced (Clauset et al. (2004b)), the measure is defined as:

$$Q = \sum_{c=1}^{n} \left[ \frac{L_c}{m} - \gamma \left( \frac{k_c}{2m} \right)^2 \right]$$

where we iterate over all $c$ communities. $m$ is the numbed of network edges, $L_c$ is the number of within-community links for specific community $c$, $k_c$ is the sum of degrees of the nodes in community $c$, and $\gamma$ is the resolution parameter.

# D    ADDITIONAL RESULTS

## D.1    EXPERIMENTAL SETUP WITH FUNCTIONAL CONNECTOMES

The resting state fMRI data from a single subject from the HCP1200 dataset pre-parcellated using the Kong2022 individual areal-level parcellation procedure were used in this study (Kong et al. (2021)). Models were examined under several parcellation settings, where the cortex is represented as functional connectomes resolutions ranging from 100 to 1000 nodes in increments of 100 nodes each (Kong et al. (2021)) and were evaluated on tasks. FC and adjacency matrix connections on each resolution describe the Pearson correlation between BOLD time series among brain regions. Regional labels and indices were collected from atlas mapping of the Schaefer atlas directory (Schaefer et al. (2017)), segregating every connectome resolution available of Kong's parcellation into 17 functional Yeo functional networks (Thomas Yeo et al. (2011)): Visual (A, B, and C), Auditory, Somatomotor (SomMotA and B), Language, Salience/Ventral Attention (SalVenAttn A and B), Control (ContA, B, and C), Default (A, B, and C), and Dorsal Attention (DorsAttnA and B).

For a weighted rs-fMRI graph we construct several thresholded graphs on different thresholding values r ranging from -0.5 to 0.4 in increments of 0.1, with a threshold value of 0 effectively keeping only positively correlated connections on the original graph. Varying r effectively filters the graph, sparsifying graph connections based on the degree of thresholding in large degrees (magnitudes of connections filtered between two consecutive thresholds). Keeping negative correlations on lower threshold values may dilute the computation graph, and potentially decrease the model's performance on several metrics. Additionally, each connectome is scaled by an alpha parameter between 0.05 and 2, which determines the value of the largest eigenvector of the connectomic adjacency matrix. The alpha scaling is applied after the normalization of all connections in the graph, and the effect of the alpha parameter on reservoir network dynamics is discussed in-depth in Suarez et al. (2020).

All models are evaluated on a perceptual decision-making task with delayed responsive, measuring both the computation and memory capacity of the reservoir networks (Inagaki et al. (2019)).

## D.2    RESERVOIR SIZE

In scaling experiments, the model generally performs better when larger parcellation sizes are used. Visualizations and numerical results indicated diminishing returns over later larger parcellations, while there is a substantial difference between the performance of the best and the worst alpha scaling values. Size efficiency decreases as resolution increases with diminishing returns starting at 400 nodes parcellation (Figure 15). On the other hand, SC reservoir with 1000 nodes has high performance at 0.82 F1-score while FC at 700 nodes already reaches 0.75 F1-score, demonstrating FC's parameter efficiency and signifying that FCs can be better scaled to finer resolutions with higher model performance.

## D.3    FUNCTIONAL THRESHOLDINGS

Hyperparameter combination experiments showed that functional connectome benefits greatly when positively thresholded, possibly confirming the hypothesis that negative correlations dampen reservoir dynamics. Reservoirs performed optimally when thresholded at 0-0.2 and scaled at alpha 0.85-1.15 (Figure 16).

## D.4    PERMUTATIONS

Input-output permutation experiments significantly varied performance on different pairings. There are several different configurations where the model performed optimally for different parcellations; however, better performance depends on the set of output nodes with Control and Default outputs outperforming other groups. Most notable are readouts from Auditory and Language functional

networks where performance is significantly worse than other readouts. Said two functional network groups have substantially fewer nodes than the other functional network groups, possibly indicating that performance heavily depends on the size of the readouts up to the optimal signal-to-noise ratio (Figure 17).

## D.5  READOUT SIZE

Readout balancing and permutation experiments show a potential correlation between the readout functional network size and the model's performance. While equal readout functional networks perform roughly uniformly, the augmented functional network readouts perform slightly better with increasing size relative to other readout functional networks. Such is the case for the Language readouts from functional connectome and DA readouts from structural connectome, both of which did not perform well on the original connectomes. However, despite the general increase in performance and only slight variance between functional network readouts, certain combinations or networks performed worse than others regardless of readout partitioning. The performance anomaly refers to the darker 2-3 performance spots in the functional connectome, and similarly in structural connectome.

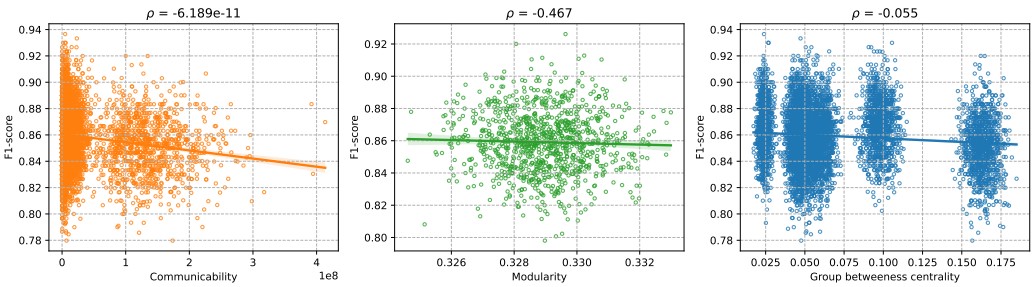

Figure 10: Structural 500 nodes, statistic-performance correlations, PDM, $\alpha = 0.95$.

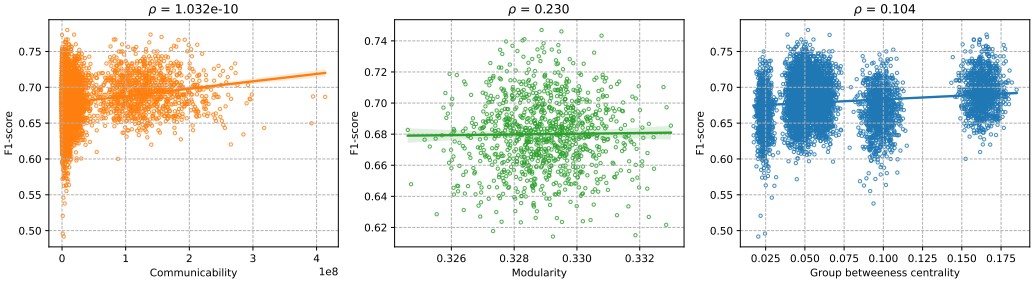

Figure 11: Structural 500 nodes, statistic-performance correlations, CDM, $\alpha = 0.95$.

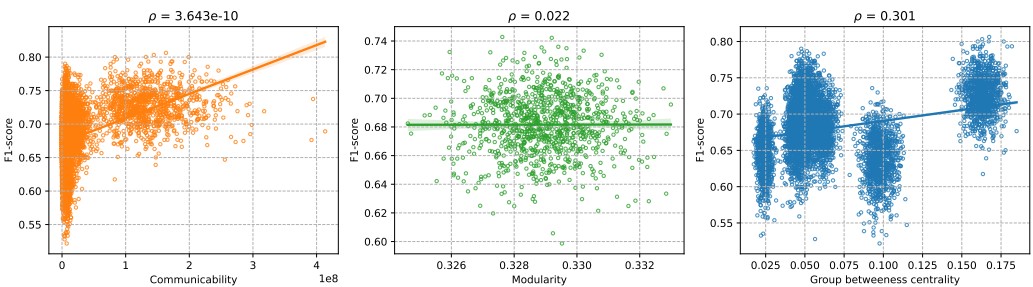

Figure 12: Structural 500 nodes, statistic-performance correlations, CDM, $\alpha = 1.2$.

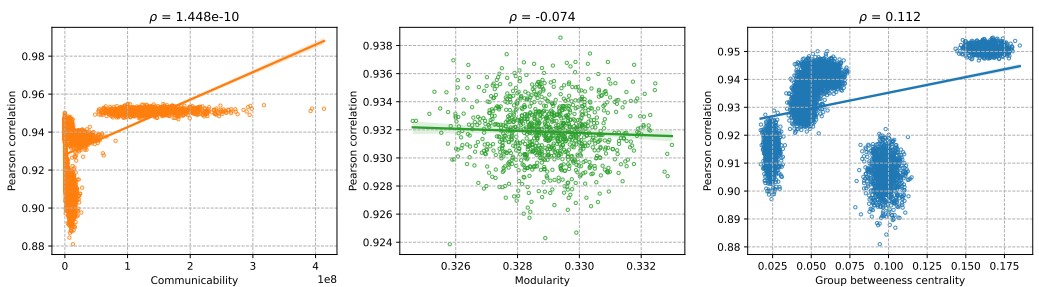

Figure 13: Structural 500 nodes, statistic-performance correlations, MemCap, $\alpha = 0.95$.

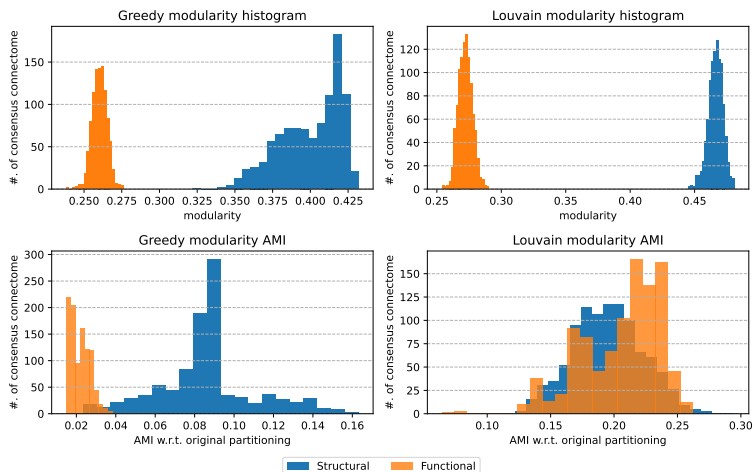

Figure 14: Modularity histogram: partition algorithm greedy modularity or Louvain modularity-based, 20 bins, with adjusted mutual information w.r.t. the original partition.

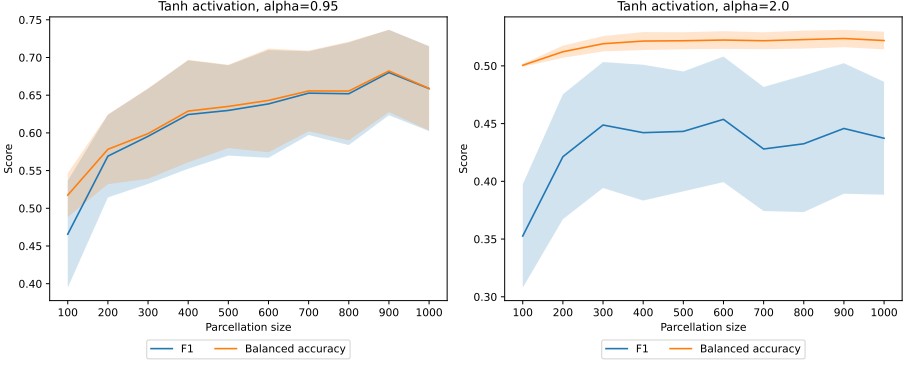

Figure 15: Performance on tanh activation on the best (**left**) and worst (**right**) alpha values.

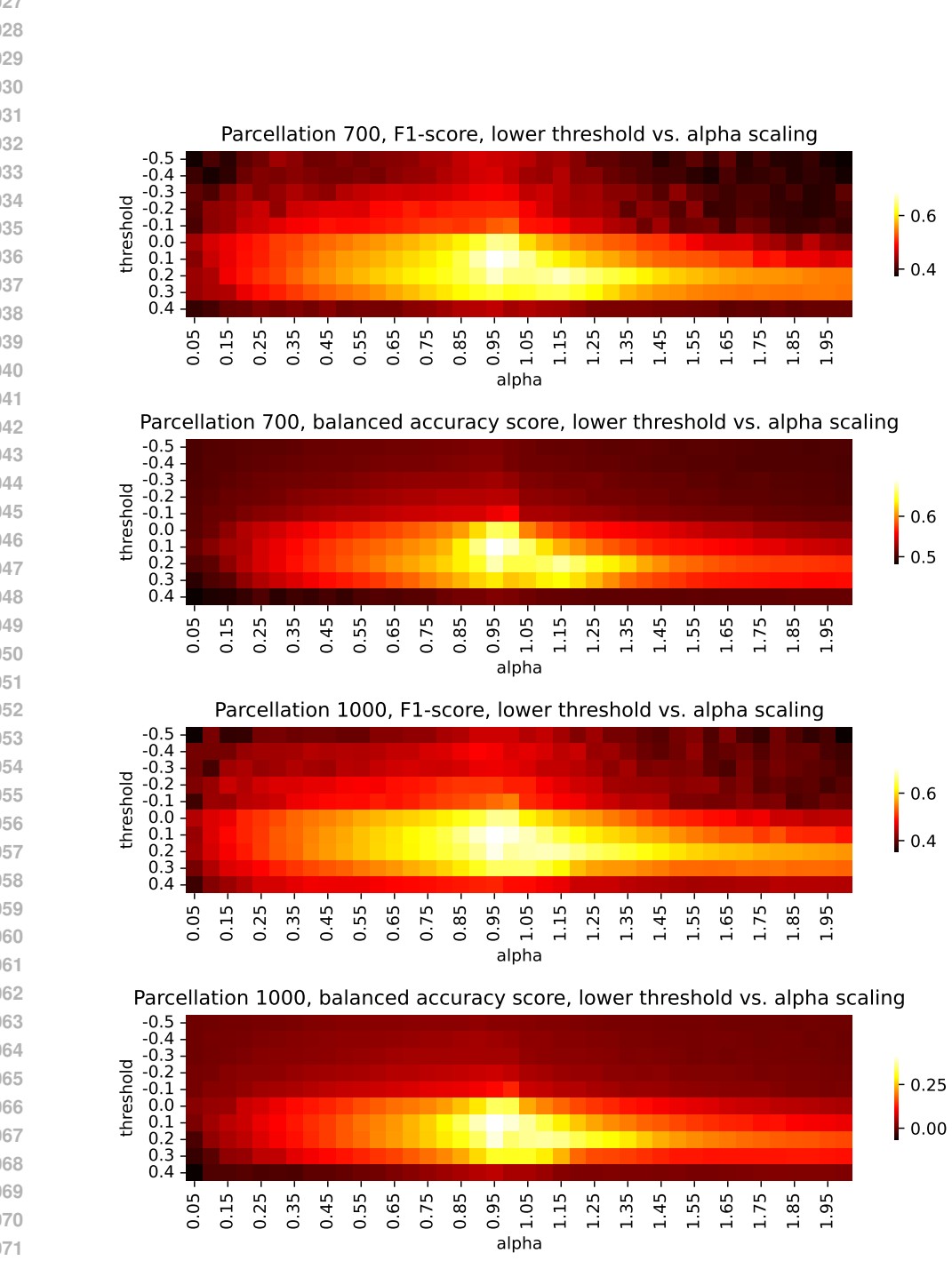

Figure 16: **Model performance w.r.t. two hyperparameters, threshold value and alpha scaling, on two notable resolutions and performance metrics.**

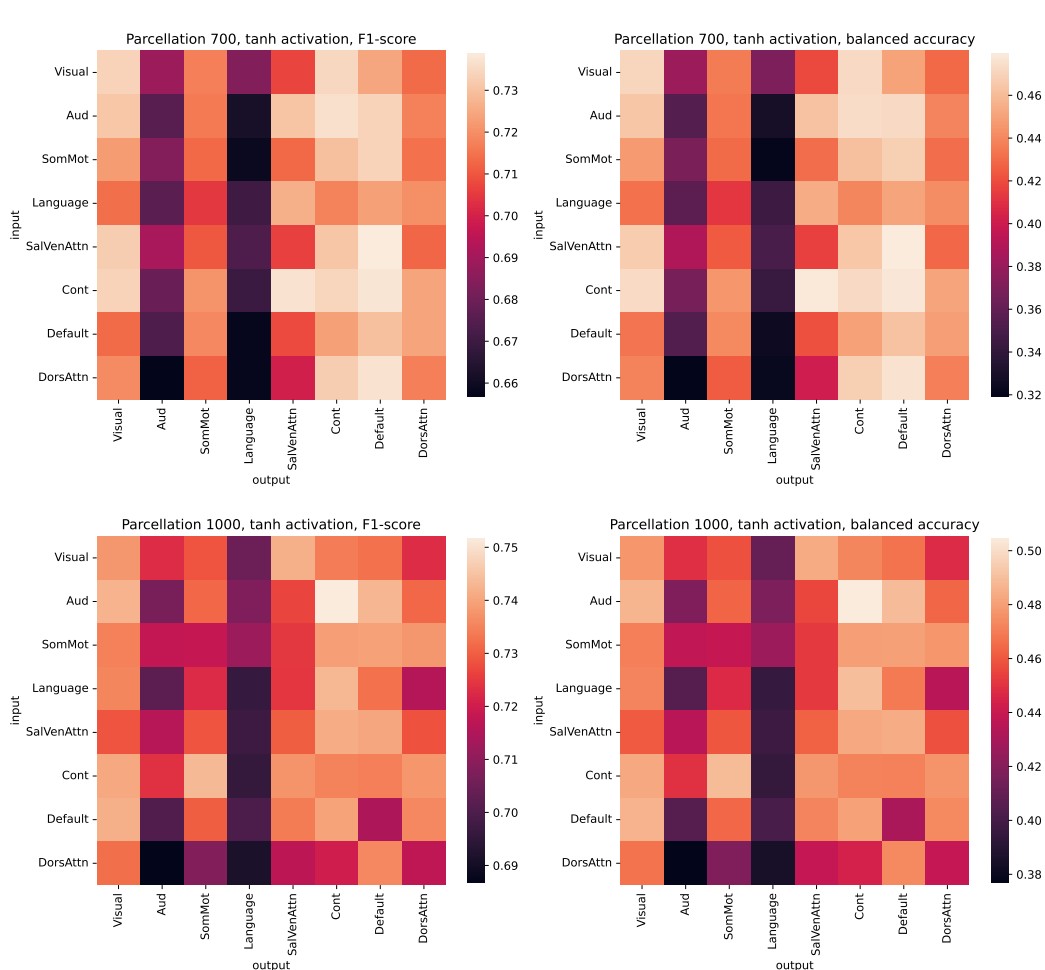

Figure 17: **Model performance w.r.t. various input-output functional network groups on 700 and 1000 parcellations, F1-score and balanced accuracy. Input** denotes the set of nodes of a certain functional network group where external stimuli representations can be routed into the reservoir. **Output** denotes the set of nodes where internal reservoir dynamics representations are read out to generate the final network results.

