# OpenReview forum: "Evaluating topological fitness of human brain-inspired sub-circuits in Echo State Networks"
_ICLR.cc/2025/Conference — ICLR 2025 Conference Withdrawn Submission_

### Official Review · Reviewer_k5To · 2024-10-16

**Soundness:** 2
**Presentation:** 2
**Contribution:** 1
**Rating:** 3
**Confidence:** 5

**Summary:**

The paper presents a already seen approach that integrates human brain-inspired topologies into ESNs. The findings suggest that neuro-physiological features of brain sub-circuits can, in some cases, improve performance in tasks related to processing and memory.

Overall the paper lack real output from ESN and reads more like a biology paper in structure, vocabulary and the nature of many big theory mistakes.
* the wrong citation for memory capacity and confusion on echo state property.
* The overall structure and vocabulary is a bit confusing.

The results presented are too weak to be considered significant. The conclusion that “different topological properties are conducive to the processing of different tasks” highlights the lack of substantial findings.

In my view, the focus of this paper should have been on the various “variants” used to generate the network rather than on ESN performance and evaluate the performance on well motivated task that resemble the ones the investigated circuit might perform. Unfortunately, the performance of the investigated connectivity models is not particularly compelling compared to traditional ESN models.

**Strengths:**

The subject is interesting but not well motivated. See questions.

A clear strength of the paper is thorough in the citation of similar works, they are well cited and the difference with present work is clear. The paper studies many variants of connectivity generations which is always good.

**Weaknesses:**

One of the claims in the paper is that it “found that the performance of the reservoir model is not strictly defined by the reservoir’s echo-state property,” but it incorrectly associates this with “the spectral radius of the connectivity matrix.” This is a misunderstanding, as it is commonly stated that if the spectral radius of the connectivity matrix is below 1, the network respects the echo-state property. However, a network can still respect the echo-state property even if its spectral radius exceeds 1. This confusion undermines one of the paper’s key claims.

Additionally, the concept of memory capacity is incorrectly attributed to Suarez et al. (2020), when it should be credited to Jaeger (2002). The paper also misuses this concept. To clarify, Suarez (2020) does cite Jaeger (2002), so this mistake is not theirs.

The structure and wording of the paper is also confusing. For instance, the section titled “Structural and functional connectomes” should not be part of the dataset description, but rather included in the section explaining how the models are formed. Referring to a model as "control" is not common in Machine learning.

The results are too weak to be really interesting as many previous paper (rightfully cited in the present one) have already explored this path.

**Questions:**

How does the temporal scale of the biological networks studied compare to the temporal scale of the tasks performed? Is there any motivation for why the temporal scale of these biological networks and the computations they perform would be relevant for the tasks being investigated?

Why does the memory capacity task use Pearson correlation as the evaluation metric?

What motivates the use of the F1-score instead of accuracy in this study?

---

### Official Review · Reviewer_LQT2 · 2024-10-31

**Soundness:** 2
**Presentation:** 2
**Contribution:** 2
**Rating:** 3
**Confidence:** 4

**Summary:**

This paper explores the impact of embedding human brain-inspired topologies, particularly the Default Mode Network (DMN), into Echo State Networks (ESNs) to enhance performance in cognitive tasks. By incorporating the structural properties of the DMN, such as modularity and high centrality, into ESNs, the study investigates whether these biologically inspired architectures can improve network performance in tasks requiring memory and decision-making. The key contribution lies in demonstrating that DMN-based ESNs outperform random networks on several tasks, suggesting that brain-inspired functional topologies can enhance artificial network performance. This work bridges neuroscience and machine learning, providing insights into how brain-like structures can be beneficial in artificial cognitive processing.

**Strengths:**

1. The study innovatively embeds human brain functional topologies, like the DMN, into ESNs, bridging neuroscience and machine learning to enhance network performance.

2  The paper is well-written, with clear descriptions of the ESN model and results, making it accessible for readers in both neuroscience and AI.

3  This work is valuable for AI and neuroscience, suggesting that brain-inspired network structures could lead to more efficient artificial networks for cognitive tasks.

**Weaknesses:**

1 The Default Mode Network (DMN) is a resting-state network and may not optimally support task-specific performance enhancements. To address this, the authors could include a rationale for using DMN in active task contexts or, preferably, conduct comparisons with task-relevant networks like the Executive or Attention Networks. Testing these would clarify if the observed performance gains are unique to DMN topology or simply due to network modularity.

2 The paper lacks comparisons with empirical neurobiological data, which would strengthen the biological plausibility of embedding functional networks in ESNs. Incorporating neural data, such as functional MRI or EEG data for verification, or comparing modeled task performance with observed human data would improve the paper's rigor and relevance to neuroscience.

3 The effects of key parameters (e.g., connection density, node centrality) on ESN performance are not fully explored. Conducting sensitivity analyses or providing ablation studies on these parameters could help understand which aspects of DMN topology are crucial for performance improvements.

4 The study only compares DMN-based ESNs with random networks, without considering other biologically plausible alternatives like small-world or scale-free networks. Including these structures would provide a broader understanding of which network topologies, beyond DMN, contribute to the observed performance gains.

5 While the study highlights improvements on experimental tasks, its implications for real-world or long-term applications remain unclear. Testing the ESN on more complex, real-world-inspired tasks or applications would offer insights into the practical relevance and robustness of the proposed approach.

**Questions:**

1 Since the Default Mode Network (DMN) is primarily a resting-state network, why was it chosen to enhance task performance, given that other task-relevant networks might be better suited?
2 Did the study explore other biologically plausible network topologies (e.g., small-world or scale-free networks) in the ESN structure?
3 Was empirical neurobiological data, such as functional MRI or EEG, considered for validating the model’s assumptions or findings?
4 Are there plans to evaluate the ESN with DMN topology on more complex or real-world-inspired tasks to assess its practical utility?
5 How sensitive is the model's performance to variations in parameters like connection density or node centrality within the DMN structure?
6 Could the authors expand on how specific DMN properties, such as modularity or centrality, might contribute to ESN performance, even beyond cognitive tasks?

**Details Of Ethics Concerns:**

no concerns

---

### Official Review · Reviewer_kfqk · 2024-11-02

**Soundness:** 3
**Presentation:** 3
**Contribution:** 3
**Rating:** 5
**Confidence:** 5

**Summary:**

In this work, the authors evaluate the computational efficiency of brain connectome with the echo state networks. Interesting, they found that that the brain topology induced some benefit in different tasks. Overall, the exploration on such interdisciplinary field is very interesting and would be inspiring for future work.

**Strengths:**

1. The approach of understanding brain topology with ESN is very interesting although I cannot exactly judge whether there are similar approaches before.

2. The paper is clearly written and easy to understand.

3. They provide comparisons on different setups.

**Weaknesses:**

1. There are some missing related works. For example, Ref[1] performs a similar analysis with RNN, Ref[2] talks about modularity with HebbFF, and Ref[3] summarizes the potential paradigms of integrating connectome to ANNs. Relating the current work with existing ones will help position the value of the current work.

2. Why do you evaluate according to the 7 readout Yeo network? Although this one is popular, it needs verification about the robustness of the results here with other potential partitions. In addition, considering the number of neurons in different networks are distinct, it is necessary to control the covariants here.

3. Many of the results, including but not limited to those in Figure 3,4,5 and Table 1 are reported with the statistical significance. Thus it is hard to tell whether the differences are statistically meaningful. Some perturbation and bootstrap based statistics should be reported here to improve the quality of the results.

4. The results in Table 1 is quite isolated with the others. The authors should focus on the cognitive tasks and provide some mechanistic insight on what kind of specific topology or property in the connectome leads to the improvement of model performance.

5. The relationship between network measurement and model performance could be investigated across subjects as well where you can adopt the FC/SC for each subject separately and evaluate their performances on tasks. HCP or other dataset with task data would be helpful here.


Ref:
[1] https://www.nature.com/articles/s41467-022-28323-7
[2] https://www.science.org/doi/full/10.1126/sciadv.adm8430
[3] https://www.nature.com/articles/s41586-023-06670-9

**Questions:**

Please check the weakness.

---

### Official Review · Reviewer_JBRq · 2024-11-04

**Soundness:** 3
**Presentation:** 3
**Contribution:** 2
**Rating:** 5
**Confidence:** 2

**Summary:**

This paper presents a method to evaluate the topological fitness of human brain-inspired echo state networks (ESNs) by embedding structural connectomes (SC) and functional connectomes (FC). It aims to assess the impact of various functional sub-circuits (derived from fMRI) on ESN performance in processing and memory tasks. Experimental results suggest that ESN performance is significantly influenced by the neurophysiological characteristics of specific functional sub-circuits across different tasks.

**Strengths:**

The paper provides empirical evidence that the topology of different sub-circuits can influence network performance, contributing a valuable experimental perspective to the study of biologically inspired network structures in computational neuroscience.

**Weaknesses:**

The contribution of this paper feels limited, as it neither introduces novel methodological advancements nor offers in-depth insights into the experimental results. The study primarily involves modifying network structure, observing performance variations, and drawing conclusions. Given the study's focus on computational neuroscience, providing theoretical explanations or a mathematical basis for the observed phenomena would be essential to strengthen the findings and their broader applicability.

**Questions:**

I recommend that the authors include theoretical and mathematical analysis to substantiate the results derived from their computational model. This would help clarify the mechanisms driving the observed impacts of different sub-circuit topologies.

---

### Note · Authors · 2024-11-30

I have read and agree with the venue's withdrawal policy on behalf of myself and my co-authors.